# Practices and outcomes of responsive caregiving on child neurodevelopment and mental health across diverse global populations: a scoping review protocol

Eunice Lobo [1,2,3] Sandeep Mahapatra [4] Giridhara Rathnaiah Babu [5]
Onno CP van Schayck,[3] Prashanth Nuggehalli Srinivas [2]
Debarati Mukherjee [1]

[1]Indian Institute of Public Health – Bengaluru, Public Health Foundation of India, Bangalore, Karnataka, India
[2]Institute of Public Health Bengaluru, Bangalore, Karnataka, India
[3]Department of Family Medicine, Care and Public Health Research Institute, Maastricht University, Maastricht, Netherlands
[4]Independent consultant, Bangalore, Karnataka, India
[5]Department of Population Medicine, College of Medicine, QU Health, Qatar University, Doha, Qatar

**Correspondence to**
Dr Debarati Mukherjee;
debarati.mukherjee@phfi.org

## ABSTRACT

**Introduction** Responsive caregiving (RC) leads to positive outcomes in children, including secure attachment with caregivers, emotional regulation, positive social interactions and cognitive development. Through our scoping review, we aim to summarise the practices and outcomes of RC in diverse caregiver and child populations from 0 to 8 years.

**Methods and analysis** We will use the Arksey and O'Malley framework and the Joanna Briggs Institute methodology for scoping reviews. We shall present our findings as per the Preferred Reporting Items for Systematic Reviews and Meta-Analyses guidelines for scoping review. Only peer-reviewed, English-language articles from 1982 to 2022 will be included from PubMed, Web of Science, APA PsychInfo, APA PsycArticles, SocINDEX and Google Scholar databases. Reference lists of included articles will also be screened. The search strategy will be developed for each database, and search results will be imported into Rayyan. Screening will be done in two phases: (1) titles and abstracts will be screened by two authors and conflicts will be resolved by mutual discussion between both or by consulting with a senior author; and (2) full-texts of shortlisted studies from the first phase will then be screened using the same inclusion/exclusion criteria. A data extraction form will be developed to collate relevant information from the final list of included articles. This form will be pilot tested on the first 10 papers and iteratively refined prior to data extraction from the remaining articles. Results will be presented in figures, tables and a narrative summary.

**Ethics and dissemination** No ethics approval needed as the review shall only use already published data. We shall publish the review in an open-access, peer-reviewed journal and disseminate through newsletters, social media pages, and presentations to relevant audiences.

## STRENGTHS AND LIMITATIONS OF THIS STUDY

⇒ Our study will follow the guidance of the Arksey and O'Malley framework and the Joanna Briggs Institute methodological framework for conducting a scoping review.

⇒ The search strategy will include databases such as PubMed, Web of Science, APA PsychInfo, APA PsycArticles, SocINDEX, along with supplementary search in Google Scholar, and cross-referencing of the reference list of included studies.

⇒ We plan to use a mixed-methods approach in this review, including both quantitative and qualitative studies to include the rich data available from both types of enquiries.

⇒ Our study is limited by language bias due to inclusion of publications in English language only, thus incurring a bias that certain studies published in other languages will not be included in our review.

## INTRODUCTION

An estimated 250 million children in low-income and middle-income countries fail to attain their developmental potential due to a range of adversities experienced during early childhood.[1] The enduring impact of these early adversities leads to major losses in human capital and productivity, thus creating a vicious cycle of intergenerational poverty. The Nurturing Care Framework launched in 2018 was prepared as a road map for caregivers, governments, civil societies, academics and other actors.[2 3] The framework includes five components that are deemed necessary for optimising child developmental potential. These include nutrition, health, responsive caregiving (RC), opportunities for play, and safety and security. Considered a foundational component, RC is defined as the ability of a caregiver to observe, notice cues (nonverbal or verbal) and respond in a timely, sensitive and appropriate manner.[4 5] This entails a back-and-forth interaction, which is also described as a serve-and-return interaction that has been known to positively shape brain architecture.[6 7] RC is rooted in sensitive responsiveness (or sensitivity) as explained by Ainsworth *et al* in 1978,[8] where the caregiver



responds to the child's implicit behavioural signals through appropriate, sensitive and contingent eye-gazes, facial expressions, positive affect, vocal responses and touch. The needs of the child are responded to with empathy and timeliness while recognising that each child has unique needs and preferences. The concept of back-and-forth interactions (serve-and-return)—the most efficient learning strategy for children—also stems from this concept of sensitive responsiveness. Research highlights the positive impact of caregivers who consistently engage in sensitive responsiveness, contributing to fostering secure attachment,[9][10] social and emotional well-being,[11] and positive developmental outcomes for children.[12][13]

RC is bidirectional in nature and requires caregiver-child interactions that form basis of secure attachment, foster emotional bonding, and develop communication and social skills. The caregiver also provides the child with an enabling environment for learning that is safe and allows the child to explore and practice learnings. Beginning from infancy, the importance of stimulation and responsive interaction, including smiling, touching, talking, singing, reading and playing, helps to develop and foster secure attachment.[14–17] For the purpose of our review, we will operationalise RC as defined by Black and Aboud in 2011, which states that RC must include behaviour that is prompt, contingent, emotionally supportive and developmentally appropriate (not intrusive) towards the child.[18]

Evidences across the years have highlighted the positive effects of RC and child outcomes. As seen in the systematic review by Aboud and Yousafzai that highlighted 21 interventions centred around responsive stimulation, it had a medium effect on children's cognitive and language development.[19] Further, a meta-analysis in 2019 of 37 studies showed that RC was significantly associated with children's language outcomes.[12] Studies across the world over the years have also shown that RC in early years leads to improved physical, cognitive and psychosocial health in children.[20–24] Moreover, data from longitudinal birth cohorts showed that RC and early opportunities of learning were linked to higher levels of cognitive abilities during adolescence in Brazil and South Africa.[25]

RC requires continuous and reciprocal interaction among the caregiver and child. Hence, characteristics and contexts including maternal age, depression, income level, along with the child's development and even gender play crucial roles.[26–29] Recently, a rapid review reported how RC was reportedly reduced during the COVID-19 pandemic, with an increase in harsh parenting due to caregiver stress, anxiety and depression.[30]

Even though there exists evidence that shows the differences in diverse populations, most of the studies and reviews are from the global north. As far as our current understanding extends, no comprehensive scoping review has been documented that describes the practices and outcomes of RC, especially based on caregiver and child characteristics and contextual factors. Given this prevailing knowledge gap, we will follow a systematic and

rigorous process, and undertake this effort to conduct an exploratory study that maps the existing literature to exclusively focus on RC practices and outcomes. Hence, a key strength of this review will be the exploration of RC practices in low-income and middle-income countries and understanding their effects on child development outcomes, a context that is under-represented in global epidemiological studies.

## METHODS AND ANALYSIS
The scoping review will be conducted using the guidance of the Arksey and O'Malley framework[31] along with the Joanna Briggs Institute method.[32] The framework will include the following steps: (1) identifying the research question/s; (2) identifying relevant studies; (3) study selection; (4) charting the data and (5) collating, summarising and reporting the results. We will describe each stage in detail below.

Additionally, we will use the Preferred Reporting Items for Systematic Review and Meta-Analysis for scoping review (PRISMA-ScR) checklist[33] for reporting. The protocol has not been registered with PROSPERO, since it does not accept scoping review protocols. Any deviations or modifications from the protocol relevant to the study will be reported in the publication of the final report.

### Stage 1: identifying the research questions
1. What do we know about RC practices across the world in terms of
   a. caregiver and child characteristics
   b. income and location settings
   c. cultural differences
   d. developmental progression over the years.
2. How do RC practices affect neurodevelopment and mental health outcomes in children (0–8 years)?

### Stage 2: identifying relevant studies
#### Search methods—information sources and search strategy
A systematic search of literature published between 1982 and 2022 will be conducted among the following databases: PubMed, Web of Science, APA PsychInfo, APA PsycArticles and SocINDEX. We will use a search strategy that includes relevant terms, medical subject headings (MeSH) with appropriate Boolean operators. (See box 1 for the pilot search strategy in PubMed.) A supplementary search in Google Scholar will also be conducted to ensure no relevant studies as missed. Additionally, cross-referencing of the reference list of the studies included will also be done, while reviews (systematic, narrative, scoping), editorials, and commentaries will not be included in the current review. Detailed documentation of every search in the aforementioned databases will be done to include information on keywords used, and number of hits or retrievable studies.

## Box 1 Pilot search strategy in PubMed database

("responsive car*"[(tiab)) OR Responsiveness[(tiab)) OR "effective parenting"[(tiab)) OR "caregiver child relation*"[(tiab)) OR "parent child relation*"[(tiab)) OR "parental responsivity"[ tiab] OR "maternal behaviour*"[(tiab)) OR "paternal behaviour*"[(tiab)) OR Mother-Child Relations*[(MeSH)) OR Mother-Child Relations*[(tiab)))
AND
(infant [(MeSH)) OR child[(MeSH)) OR baby[(tiab)) OR babies[(tiab)) OR toddler*[(tiab)) OR preschool*[(tiab)) OR pre school*[(tiab)) OR "young child*"[(tiab)) OR "early childhood"[(tiab)) OR infan*[(tiab)) OR new-born*[(tiab)) OR new-born*[(tiab)) OR minor*[(tiab)) OR kid*[(tiab)) OR child*[(tiab)) OR pediatricpaediatric*[(tiab)) OR pediatricpaediatric*[(MeSH)))
AND
(neurodevelop*[(tiab)) OR "mental health"[(tiab)) OR cognit*[(tiab)) OR "Mental health"[(MeSH)) OR "Child Development"[(tiab)) OR "infant development"[(tiab)) OR "Child BehaviorBehaviour"[(MeSH)) OR "Child Development"[(MeSH)) OR "Cognition"[(MeSH)))

### Stage 3: study selection based on inclusion and exclusion criteria

Table 1 summarises the details of exclusion and inclusion criteria of studies.

### Types of study

Our review will include published research articles in English language.

### Population

For our review, we will include studies conducted among young children (age: 0–8 years), and their caregivers, including biological parents, alternate caregivers (grandparents, siblings, aunts), foster, adoptive parents, etc. Since early childhood period, that is, the time from birth to 8 years, is a critical period in the development of many foundational skills in all areas of development,

and RC has been considered the foundational component of nurturing care, with evidence supporting positive outcomes associated with early childhood development, we have included the same age range for children.

### Concept

The studies included will describe practices of RC; hence, we will include studies that mention different forms of RC. The most commonly used terms are responsiveness, sensitivity, caregiver–child interaction, etc. Based on preliminary literature search, additional relevant terms will be added to develop the search string.

### Context

Since our aim is to synthesise a comprehensive review of RC among diverse populations, we will include studies from low-income as well as high-income settings based on the socioeconomic status mentioned in the individual studies, papers from urban, rural locations, global south and north, different types of families/caregivers, cultures, etc. We will also consider studies reporting data from children with atypical development, disabilities and injuries.

### Selection of studies

Search results from the databases will be imported in the Rayyan software,[34] and removal of duplicate articles shall be carried out for first level screening. Based on our inclusion and exclusion criteria, articles will be screened for review by two independent authors (EL, SM) by screening the title and abstracts through a blind review, that is, each author will be kept blind to the decision to 'include' or 'exclude' articles. This will be followed by a full-text review of articles that are available. Any and all discrepancies among the authors at each step shall be discussed and resolved by senior authors (DM, PNS). At the end of the screening phase, the inter-rater agreement between

| Table 1 | Eligibility criteria for study selection | |
|---|---|---|
| **Eligibility criteria** | **Inclusion criteria** | **Exclusion criteria** |
| Source of evidence | Electronic databases, reference list of included studies and Google Scholar | |
| Population | Children ≤8 years<br>Caregivers, including biological parents, alternate caregivers (grandparents, siblings, aunts), foster, day-care providers and adoptive parents. | |
| Concept | Responsive caregiving (practices, differences, stimulation, feeding, outcomes) | |
| Context | Diverse populations including caregiver, child characteristics, income, location, cultural differences | |
| Publication status | Published peer-reviewed studies | |
| Language | English | Publications in other languages such as French, Chinese, etc. |
| Study designs | Primary study designs such as quantitative including observational, randomised controlled trials, interventions, qualitative and mix-method studies with human participants | Non-human/Animal studies; editorials and commentaries; other review studies such as systematic, narrative and scoping reviews |

the reviewers will be computed using the Cohen's kappa coefficient (κ) statistic.[35]

## Stage 4: charting the data

An online form will be developed for the data extraction and will include the following details: (1) author; (2) year of publication; (3) title; (4) aims and/or objective or research question; (5) country of study; (6) study design; (7) study participants characteristics, that is, child and caregiver details such as age, sex, education, occupation, income status; and (8) study result/s relevant to our research question/s—including RC practices and outcomes. The form will be initially piloted by two authors (EL, SM) with 10 included studies to ensure accuracy in terms of the study objectives and will be modified and revised as necessary. The data extraction will be done independently by the same authors, and disagreements or differences of opinions will be resolved through mutual discussion or need be by a third, senior author (DM). Through the continuous data extraction process, authors will ensure all relevant data shall be captured, and all authors are aware of the progress. All changes or modifications will be explained in the review publication.

## Stage 5: collating, summarising, and reporting the results

For the synthesis of results of our scoping review, we will prepare a PRISMA flowchart describing the search and screening of articles according to the aforementioned PRISMA-ScR checklist. This will be followed by a detailed report prepared after analysing the extracted study characteristics. We will also include qualitative studies; hence, thematic analysis of the qualitative research articles will be done and included. Therefore, based on the research questions developed, the review will include a meticulous and comprehensive understanding of RC practices and outcomes across diverse populations across the world.

## Patient and public involvement

No patients were involved in the protocol design.

## DISCUSSION

Our proposed scoping review will aim to describe RC across global settings, that is, global south as well as global north with a special focus on whether differences exist among diverse populations. To the best of our knowledge, this review will be the first to inform regarding the practices of RC across the world. Thus, we aim to understand characteristics of caregivers and children as well as context-specific practices. We will also aim to learn about outcomes of RC on child development during the age of 0–8 years. We believe our review will add value to the existing knowledge by consolidating the information as well as drawing attention to the gaps in practice of RC and early childhood development. Hence, it will hold implications for policy, practice and research aimed at enhancing the quality of caregiving practices and

subsequently positively impacting child developmental outcomes worldwide.

## Ethics consideration and dissemination

Ethics approval shall not be required as this study will retrieve and synthesise data from available published literature. The scoping review will be published in an open-access, peer-reviewed journal with a high impact factor to reach larger audiences and outreach to public health researchers, social scientists, early childhood development experts, health services researchers etc. Our review findings shall be additionally presented via our study newsletter, social media pages related to the study, as well as institute seminars and journal clubs.

**Contributors** All authors have made substantive intellectual contributions to the development of this protocol. EL, DM and PNS conceptualised the idea and the research questions. EL and SM developed and tested search terms. EL drafted the first draft of the manuscript, which was then followed by numerous iterations with critical and substantial input and appraisal from all of the authors (SM, GRB, OVS, PNS, DM). All authors approved the final version of the manuscript.

**Funding** This work is supported by the DBT/Wellcome Trust India Alliance Team Science Grant [Grant No. IA/TSG/20/1/600023] (Awarded to GRB as lead Principal investigator). Prashanth N Srinivas was supported by the DBT/Wellcome Trust India Alliance CRC Grant [Grant IA/CRC/20/1/600007] awarded to him. The funders had no role in study design, methods, decision to publish, or preparation of the manuscript.

**Competing interests** None declared.

**Patient and public involvement** Patients and/or the public were not involved in the design, conduct, reporting, or dissemination plans of this research.

**Patient consent for publication** Not applicable.

**Provenance and peer review** Not commissioned; externally peer reviewed.

**ORCID iDs**
Eunice Lobo http://orcid.org/0000-0002-0621-9295
Sandeep Mahapatra http://orcid.org/0000-0002-7544-8382
Giridhara Rathnaiah Babu http://orcid.org/0000-0003-4370-8933
Prashanth Nuggehalli Srinivas http://orcid.org/0000-0003-0968-0826
Debarati Mukherjee http://orcid.org/0000-0003-3473-6537

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
