## [Reviewer comments · BMJ Open]

ARTICLE DETAILS

TITLE (PROVISIONAL)	Practices and outcomes of responsive caregiving on child neurodevelopment and mental health across diverse global populations: A scoping review protocol
AUTHORS	Lobo, Eunice; Mahapatra, Sandeep; Babu, Giridhara R; van Schayck, Onno; Prashanth, NS; Mukherjee, Debarati

VERSION 1 – REVIEW

REVIEWER	Keerty Nakray Centre for Ethics , Yenepoya (Deemed to be University)
REVIEW RETURNED	06-Nov-2023

GENERAL COMMENTS	The authors propose an exciting study on responsive caregiving (RC) to comprehend positive outcomes in children. They will use the Arksey and O'Malley framework and the Joanna Briggs Institute methodology for scoping reviews. They will present their findings based on the Preferred Reporting Items for Systematic Reviews and Meta-Analyses (PRISMA) guidelines for scoping review (PRISMA-ScR). 1. The authors aim to consider both quantitative and qualitative studies worldwide. It will be challenging. Will they include anthropological studies? Should the authors check out the challenges of combining mixed-methods, quantitative, and qualitative studies in their pilot study and decide to include everything in one paper or publish two articles? Also, considering the large publication time frame of the articles.2. Including some RCTs might help to understand interventions on harsh punishments or nutrition, cash-transfers (or maybe they pursue them in the future)3. The authors could search Cochrane Reviews About Cochrane Reviews Cochrane Library and Campbell Collaborations Child and Adolescent Mental Health and psychosocial support interventions: An evidence and gap map of low- and middle-income countries - The Campbell Collaboration4. What sample keywords they will use to search qualitative studies? Will they be the same quantitative studies?5. Authors should reflect on:a. Cultural and ethnic dimensions of responsive care,b. Parental dynamics in EECE (father-mother concerns) or multi-generational households and caregivingc. Euro-centric perspectives on care versus developing countries.d. Formal/Informal CareMarginalised groups such as migrants, ethnic minorities, race, caste, tribes and class differences in care.
---

REVIEWER	Elizabeth Hentschel Harvard University T H Chan School of Public Health, Global Health and Population
REVIEW RETURNED	27-Dec-2023

GENERAL COMMENTS	This protocol is extremely timely, very little is known about responsive caregiving behaviors in non-Western, rural settings, and the field needs a comprehensive review of existing research on responsive caregiving. The mixed-methods approach is also novel, as qualitative evidence is needed to truly understand the variation in the types of responsive caregiving used. In your introduction, I think it's worth mentioning that responsive caregiving stems from sensitive responsiveness, and give a brief overview of that literature, as this is the terminology typically used in most of Europe still. This should also be incorporated into your search strategy. Also, make sure that your definition of responsive caregiving is clear and consistent. I recommend the Black & Aboud definition or Eshel's definition. The Rasheed & Yousafzai study that you cite in the introduction measures mother-child interactions, not responsive interactions specifically. Jeong et al., conducted a scoping review in 2022 that looked at available measurement tools used to capture the components of the NCF in LMICs. A major limitation of this review was that a clear definition of responsive caregiving wasn't used, the authors included all tools that claimed to capture responsive caregiving. However, it would be worth cross-checking with this review once you conduct your full text review to make sure you aren't missing any articles that meet your inclusion criteria. What is the rationale of including children up to age 8? I'm unaware of responsive caregiving literature that works with children above the age of 5. I'm not sure if you want to include "warmth" as a term synonymous with responsive caregiving. Many psychologists today argue that warmth is not an essential component of responsive caregiving and that including it leads to conceptual confusion (see Mesman et al., 2018; Universality without Uniformity article).
--

VERSION 1 – AUTHOR RESPONSE

Reviewer: 1

Dr. Keerty Nakray, Centre for Ethics

Comments to the Author:

The authors propose an exciting study on responsive caregiving (RC) to comprehend positive outcomes in children. They will use the Arksey and O'Malley framework and the Joanna Briggs Institute methodology for scoping reviews. They will present their findings based on the Preferred Reporting Items for Systematic Reviews and Meta-Analyses (PRISMA) guidelines for scoping review (PRISMA-ScR).

Response: We thank the reviewer for their time to review our manuscript and appreciate the encouraging comments. We highly value your meticulous attention to detail and insightful comments that have enhanced the quality of our manuscript.

1) The authors aim to consider both quantitative and qualitative studies worldwide. It will be challenging. Will they include anthropological studies? Should the authors check out the challenges of combining mixed-methods, quantitative, and qualitative studies in their pilot study and decide to

include everything in one paper or publish two articles? Also, considering the large publication time frame of the articles.

Response: Thank you for your valuable suggestion regarding the challenges posed by incorporating both quantitative and qualitative studies in a scoping review. We acknowledge the complexities involved, and your recommendation regarding the pilot study and the option of publishing separate articles is well-taken. We believe that it will be feasible to summarize the information in one review article, especially since the nuances that are missed by quantitative studies but complemented by insights gleaned from qualitative studies can be discussed in the same article. It is important to note that the concept of responsive caregiving has been introduced in the literature relatively recently, and therefore, the number of studies that have been retrieved and meets our selection criteria is possible to synthesize in one article, even though the review spans the past forty years.

Study selection will strictly follow the inclusion/exclusion criteria stated in the methods. If anthropological studies retrieved from the listed databases meet all the stated criteria, then they will be included in the scoping review.

2) Including some RCTs might help to understand interventions on harsh punishments or nutrition, cash-transfers (or maybe they pursue them in the future)

Response: Thank you for your suggestion. We recognize the potential insights that Randomized Controlled Trials (RCTs) and interventions may provide, and hence have included the same in our scoping review eligibility criteria.

We thank the reviewer for highlighting this nuance in the selection criteria that we missed reporting in detail in the earlier version of the protocol. We have updated Table 2 (page 14; Main document marked copy) to reflect this inclusion criterion.

3) The authors could search Cochrane Reviews About Cochrane Reviews | Cochrane Library and Campbell Collaborations Child and Adolescent Mental Health and psychosocial support interventions: An evidence and gap map of low- and middle-income countries - The Campbell Collaboration

Response: Once again, thank you for your time and thoughtful feedback. We will thoroughly review the recommended suggestions and assess them to identify relevant articles.

4) What sample keywords they will use to search qualitative studies? Will they be the same quantitative studies?

Response: Thank you for this important question regarding the keywords for searching qualitative studies. We will use the same set of keywords for both qualitative and quantitative studies (see Table 1). Studies will be selected only on the basis of the concept of responsive caregiving, the population (children below 8 years of age), and child outcomes. Since the search strategy does not include keywords that select articles on the basis of the type of study, we anticipate both quantitative and qualitative studies to be included in the search results. However, we look forward to learning about any strategy that may be available for specifically selecting quantitative versus qualitative studies which can potentially help identify any relevant articles that our current search strategy may have missed.

5) Authors should reflect on:

- a) Cultural and ethnic dimensions of responsive care,
- b) Parental dynamics in EECE (father-mother concerns) or multi-generational households and caregiving
- c) Euro-centric perspectives on care versus developing countries.
- d) Formal/Informal Care
- e) Marginalised groups such as migrants, ethnic minorities, race, caste, tribes and class differences in care.

Response: Many thanks for your detailed reflections on the various dimensions to be considered for responsive caregiving practices globally, especially since it directly relates to the primary research

question of this scoping review. Our search strategy does not select studies based on caregiver characteristics but only on responsive caregiving practices. Therefore, any study that reports on responsive caregiving, irrespective of caregiver characteristics and/or location, will be selected into our search results. We anticipate that this will allow us to comprehensively explore responsive caregiving in diverse settings and parenting contexts by including caregiving practices across the global South and North (to address Euro-centric perspectives on care versus in developing countries), populations spanning diverse socio-economic and cultural groups (to address cultural and ethnic dimensions and marginalized groups), and different family structures (multi-generational including grandparents, siblings versus two-parent versus single-parent households) and formal versus informal caregivers. On similar lines, we have also identified studies related to responsive caregiving by mothers of differing characteristics (working versus home-makers, typical birth versus assisted methods).

We thank you for your suggestion on including the concept of cultural variations and its influence on responsive caregiving. We believe that cultural nuances play a pivotal role in responsive caregiving practices and have therefore added it as a specific dimension to explore as part of our primary research question. Please see Stage 1: Identifying the research questions on page 6 lines 131-136 - Main document marked copy; and Table 2, page 14).

Reviewer: 2

Dr. Elizabeth Hentschel, Harvard University T H Chan School of Public Health

Comments to the Author:

This protocol is extremely timely, very little is known about responsive caregiving behaviors in non-Western, rural settings, and the field needs a comprehensive review of existing research on responsive caregiving. The mixed-methods approach is also novel, as qualitative evidence is needed to truly understand the variation in the types of responsive caregiving used.

Response: Our sincere thanks to the reviewer for this encouraging comment. We appreciate the attention to detail and the suggestions to improve the quality of this manuscript.

1) In your introduction, I think it's worth mentioning that responsive caregiving stems from sensitive responsiveness, and give a brief overview of that literature, as this is the terminology typically used in most of Europe still. This should also be incorporated into your search strategy.

Response: Thank you for this important suggestion. We have edited the introduction to highlight the origins of the concept of responsive caregiving (see page 4, lines 72-82; Main document marked copy): The text now reads as,

"Responsive caregiving is rooted in sensitive responsiveness (or sensitivity) as explained by Ainsworth et al in 1978,8 where the caregiver responds to the child's implicit behavioural signals through appropriate, sensitive and contingent eye-gazes, facial expressions, positive affect, vocal responses, and touch. The needs of the child are responded to with empathy and timeliness while

recognizing that each child has unique needs and preferences. The concept of back and forth interactions (serve-and-return) - the most efficient learning strategy for children also stems from this concept of sensitive responsiveness. Research highlights the positive impact of caregivers who consistently engage in sensitive responsiveness, which contribute to fostering secure attachment,⁹ 10 social and emotional well-being,¹¹ exploration and problem-solving, and positive developmental outcomes in children.^{12 13}

Moreover, we will include the terminologies “maternal sensitivity”, “paternal sensitivity”, “parental sensitivity” in our final search strategy to capture relevant literature and contribute to a more thorough and nuanced exploration of the topic.

2) Also, make sure that your definition of responsive caregiving is clear and consistent. I recommend the Black & Aboud definition or Eshel’s definition. The Rasheed & Yousafzai study that you cite in the introduction measures mother-child interactions, not responsive interactions specifically.

Response: Thank you for your valuable suggestion. We have now included Black & Aboud (2011)[1] definition in the introduction and used it to operationalize the concept of responsive caregiving in this review.

Our text now reads as:

“For the purpose of our review we will operationalize responsive caregiving as defined by Black and Aboud in 2011 which states that responsive caregiving must include behaviour that is prompt, contingent, emotionally supportive, and developmentally appropriate (not intrusive) towards the child. 2” (see page 4, lines 88 to 91; Main document marked copy)

3) Jeong et al., conducted a scoping review in 2022 that looked at available measurement tools used to capture the components of the NCF in LMICs. A major limitation of this review was that a clear definition of responsive caregiving wasn’t used, the authors included all tools that claimed to capture responsive caregiving. However, it would be worth cross-checking with this review once you conduct your full text review to make sure you aren’t missing any articles that meet your inclusion criteria.

Response: Thank you for highlighting the scoping review conducted by Jeong et al. (2022) on measurement tools for capturing components of the Nurturing Care Framework (NCF) in LMICs. We appreciate your guidance as this is an important and related review that can inform our study. We also appreciate your attention to the definition of responsive caregiving, and as detailed in our response to your second point, we have tried to restrict study selection to only those that have specifically investigated responsive caregiving as defined by Black and Aboud [1]. As suggested, we will cross-check with Jeong et al.’s review and include any relevant articles that match our selection criteria.

4) What is the rationale of including children up to age 8? I’m unaware of responsive caregiving literature that works with children above the age of 5.

Response: We selected the age range of 0 to 8 years to align with the World Health Organization’s [2, 3] definition of early childhood development. However, in the light of your suggestion, if we are unable to identify studies that have investigated responsive caregiving in children above 5 years of age, we will highlight this as a limitation of the current literature since it is not known whether responsive caregiving continue to benefit children and adolescent outcomes beyond the first five years of life. Ainsworth has suggested that appropriateness of parental response (and therefore sensitivity) depends on the child’s developmental stage [4]. Therefore, studies beyond the first five years of life are warranted and we believe this would be an important evidence gap highlighted by our scoping review.

5) I’m not sure if you want to include “warmth” as a term synonymous with responsive caregiving. Many psychologists today argue that warmth is not an essential component of responsive caregiving and that including it leads to conceptual confusion (see Mesman et al., 2018; Universality without Uniformity article).

Response: Thank you for highlighting this important nuance and the potential for confusion about the concept of responsive caregiving by including warmth as one of the components. In fact, the definition of Black and Aboud [1] also does not include the concept of warmth, so we agree with your point. The suggested reference also makes this case very convincingly. We have removed the term “warmth” from our description of the concept of responsive caregiving (page 7, Line 168 - Main document marked copy), and will refrain from using the term “warmth” in our final search strategy as well keeping this point in mind throughout the study selection process.

VERSION 2 – REVIEW

REVIEWER	Elizabeth Hentschel Harvard University T H Chan School of Public Health, Global Health and Population
REVIEW RETURNED	25-Jan-2024
GENERAL COMMENTS	Thank you for addressing all of my comments. I am excited for the results of this review.